# Role of the NUDT Enzymes in Breast Cancer

**DOI:** 10.3390/ijms22052267

**Published:** 2021-02-25

**Authors:** Roni H. G. Wright, Miguel Beato

**Affiliations:** 1Center for Genomic Regulation (CRG), Barcelona Institute of Science and Technology (BIST), Dr. Aiguader 88, 08003 Barcelona, Spain; 2Basic Sciences Department, Faculty of Medicine and Health Sciences, Universitat Internacional de Catalunya, 08003 Barcelona, Spain; 3Department of Life Science, Universitat Pompeu Fabra (UPF), 08003 Barcelona, Spain

**Keywords:** breast cancer, drug discovery, NUDIX, PARP1, inhibitors, cancer stem cells, NUDT hydrolases, signaling pathways, hormone receptor-positive cancers, metastasis, NUDT5, MTH1

## Abstract

Despite global research efforts, breast cancer remains the leading cause of cancer death in women worldwide. The majority of these deaths are due to metastasis occurring years after the initial treatment of the primary tumor and occurs at a higher frequency in hormone receptor-positive (Estrogen and Progesterone; HR+) breast cancers. We have previously described the role of NUDT5 (Nudix-linked to moiety X-5) in HR+ breast cancer progression, specifically with regards to the growth of breast cancer stem cells (BCSCs). BCSCs are known to be the initiators of epithelial-to-mesenchyme transition (EMT), metastatic colonization, and growth. Therefore, a greater understanding of the proteins and signaling pathways involved in the metastatic process may open the door for therapeutic opportunities. In this review, we discuss the role of NUDT5 and other members of the NUDT family of enzymes in breast and other cancer types. We highlight the use of global omics data based on our recent phosphoproteomic analysis of progestin signaling pathways in breast cancer cells and how this experimental approach provides insight into novel crosstalk mechanisms for stratification and drug discovery projects aiming to treat patients with aggressive cancer.

## 1. Introduction

### 1.1. Metastatic Breast Cancer

Currently, breast cancer continues to be the leading cause of cancer death in women worldwide [1,2]. It has been suggested that 1 in 8 women will be diagnosed with breast cancer in their lifetime, and 1 in 38 will lose their lives to the disease [3]. Breast cancer is a complex disease characterized firstly as either ductal carcinoma in situ (DCIS), where the cancer has not grown out of the duct, or invasive breast cancer, which can be further characterized into either invasive ductal carcinoma (IDC) or invasive lobular carcinoma (ILC). IDC accounts for around 80% of invasive carcinomas. 

Following detection, further characterization is routinely carried out to ascertain the hormone receptor status of the tumor, which will decide whether the patient may benefit from anti-hormonal therapies. Initially, around 80% of breast cancers are estrogen receptor alpha (ERα)-positive, and from this, 65% are also progesterone receptor (PR)-positive. Treatment options for hormone receptor-positive breast cancers include selective estrogen receptor modulators (SERMs), such as tamoxifen (Novadex^TM^) or Fulvestrant (Faslodex^TM^), and aromatase inhibitors, which block estrogen production, such as anastrozole (Arimidez^TM^) and exemestane (Aromasin^TM^). In addition, 20% of breast cancers are positive for human epidermal growth factor receptor 2 (HER2) and are routinely treated with trastuzumab (Herceptin^TM^) [4,5].

Following radiotherapy, surgery, chemotherapy, and adjuvant therapy, the majority of patients become disease-free long term [6]. However, even considering this group of patients with a long-term favorable outcome, the majority of breast cancer-related deaths are due to metastasis and not the primary tumor. Although HR+ breast cancers predict a more favorable outcome at diagnosis compared to HR- tumors, the long term (>5 years) recurrence rate is significantly higher for HR+ breast cancers compared to HR-tumors, which show a peak in recurrence earlier (2 years after surgery) [7]. Recurrent disease is often incurable with 5-year survival rates less than 50%. To date, there are no diagnostic tools that reliably predict the long-term recurrence of these tumors. Moreover, very few targeted therapies exist for the treatment of metastasis, and often first line treatments are used, which are less effective. In addition, metastatic tumors commonly become resistant to adjuvant therapies [8,9]. 

The lack of treatment options for metastatic breast cancer [10], and the fact that the majority of patients will eventually relapse even if advanced disease initially responded to treatment, highlights the importance of a greater understanding of breast cancer progression. A more in-depth analysis may open new avenues for future drug discovery projects for breast cancer, which may also have translational applications for other cancer types depending upon the pathway affected. 

### 1.2. Use of 3D Cell Culture as a Model for Metastatic Breast Cancer 

Pluripotent breast cancer stem cells (BCSC) have been shown to be resistant both to chemo- and radiotherapy and are the known initiators of metastasis [11,12]. The percentages of BCSCs are relatively low in the primary tumor in vivo and in 2D cell culture *in vitro*. In recent years, the growth of breast cancer cells in non-adherent conditions has been increasingly used as a more suitable model for studying aggressive cancer cell growth [13,14,15]. In contrast to 2D cultures, the growth of breast cancer cells in non-adherent conditions as three-dimensional (3D) oncospheres leads to enrichment of BCSCs [16,17]. In fact, only these BCSCs are able to grow in non-adherent conditions and express genetic markers of epithelial to mesenchymal transition (EMT) and “stemness”, more similar to their in vivo counterparts [18,19]. Indeed, there are many different methods in order to grow cells in 3D culture. For example, in suspension in non-adherent plates, in concentrated medium in gel-like substances such as Matrigel, or on a scaffold such as silk, collagen or alginate. The advantages and disadvantages of each method have been extensively described elsewhere [15,20].

Studies that focus on understanding hormone receptor-dependent breast cancer signaling using a 3D cell culture model have shed light on several potential drug target options [21,22,23]. One such target is the enzyme NUDT5 (also known as NUDIX5, Nudix-linked to moiety X-5), which we have shown to be essential for HR+ breast cancer growth and breast cancer stem cell (BCSC) initiation and maintenance [24,25]. NUDT5 is a member of a much larger family of enzymes. In this review, we aim to summarize what is known about the role of NUDT5 and some other members of the NUDT family, in cancer growth and we will discuss how the use of global omic datasets could be integrated to provide additional insight into drug discovery opportunities for aggressive HR+ breast cancers and other cancer types. 

The NUDT/NUDIX (Nucleoside Diphosphate Linked to Moiety X) type hydrolase superfamily includes an evolutionary conserved large group of proteins, which hydrolyze a wide range of substrates playing an essential role in important biological processes including cell proliferation, signal transduction and homeostasis (Table 1). An in-depth discussion of the whole enzyme family, including reactions and structures, is not the focus of this review, as it has already been covered extensively in recent years [26,27,28].

## 2. Role of NUDT Enzymes in Cancer

### 2.1. Role of NUDT Enzymes in Breast Cancer

Analysis of breast cancer patient data reveals some interesting results stratifying patients based on the expression level of individual NUDT family members. In particular for NUDT1, 2, 5, and 16. The expression level of these four NUDT enzymes if significantly higher in tumor versus normal tissue based on data from TGCA breast cancer data (Figure 1a−d *left*). In addition, patients with elevated levels of NUDT1, 2, 5 or 16 exhibit significantly poorer overall survival in HR-positive breast cancer (Figure 1a−d *center*) [25] but not HR-negative breast cancer (Figure 1a−d *right*). This is further highlighted based on other studies where inhibition of NUDT1 (MTH1) reduced breast cancer cell growth in vitro and in vivo [29]. Similarly, NUDT2 is overexpressed in invasive ductal carcinoma, associated with poor clinical outcome [30]. Inhibition of NUDT2 shows promise as a novel chemotherapeutic target, demonstrated by a strong reduction on metastatic pathways [30,31]. 

The suggestion for an important role of this family of enzymes if further compounded by an analysis of the number of breast cancer patient datasets in which the expression level of a NUDT family member is associated with a significant effect on patient outcome (*p* < 0.05) (Figure 2a). Interestingly, NUDT4, 13 and 21 are associated with a strong effect on patient outcome in breast cancer patients (79, 29, and 46 datasets respectively). However, relatively little is known to date in the literature with regards to the role of these NUDT family members in breast cancer. Only NUDT13 was identified as up-regulated HR-positive breast cancer [32]. Low expression of NUDT21 is associated with tumor size, stage, and metastasis correlating with poor overall and recurrence-free survival in breast cancer patients, and overexpression of NUDT21 inhibits invasion, EMT, and proliferation in cell culture [33]. Further investigation into the role of NUDT4, 13, and 21 in breast cancer progression may represent an interesting line of investigation in the future. 

In recent years, the role of NUDT5 in the biology of breast cancer cells has attracted our attention in the context of our studies on the function of the poly-ADP-Ribose (PAR) synthesis by PARP1 for the hormonal gene regulation. We showed that within minutes of progesterone exposure, PARP1 is activated by CDK2-dependent phosphorylation of two serines in the active center, leading to a dramatic increase of nuclear PARylation that is essential for gene regulation [34,35]. We assumed that the increase in PAR levels will neutralize the positive charge of the histone tails weakening their interaction with DNA and resulting in a more open chromatin state [36,37,38,39,40], which should facilitate the changes in gene expression required for reprogramming gene expression [41,42,43]. Similar explanations were proposed for the role of PARylation in *Drosophila* development [44]. However, the observation that not only PAR generation but also PAR degradation to ADPR-ribose (ADPR) units by the PARG was required for the hormonal gene regulation forced us to consider other possible functions of ADPR [24]. Moreover, we observed that NUDT5 was identified in the PAR interactome of breast cancer cells exposed to hormone [24], leading us to investigate the role of NUDT5 in hormone action. We found that PARP1, PARG, and NUDT5 were all required for the displacement of linker histone H1, the first step of chromatin remodeling essential for enabling gene regulation by estrogens or progestins [24,45]. Since the remodeling of chromatin also requires ATP-dependent chromatin remodeling complexes, we considered the possibility that NUDT5 could convert ADPR to ATP by pyrophosphorylation, a reaction postulated by Sei-ichi Tanuma 24 years ago [46]. Measuring the levels of nuclear ATP with sensors of the TAP/ADP ratio, we confirmed the increase levels in cells exposed to hormones for 20–30 min hormone exposure, and found that the increase was dependent on PARP1, PARG, and NUDT5 [24]. 

NUDT5 was known to hydrolyze ADPR to AMP and Ribose-5-phosphate (R5P) [47], and we confirmed this activity using recombinant NUDT5. The crystal structure of NUDT5 shows it as a homodimer with two deep substrate cavities, each of them formed by residues from the two monomers [48]. However, in the known structure there is no possibility for pyrophosphate entering the substrate pocket. Therefore, a structural conformation change was needed. Indeed, we found that immediately after hormone exposure NUDT5 is dephosphorylated at threonine 45, leading to a change in the orientation of the two monomers in the homodimer, which enables binding of pyrophosphate and generation of ATP [24,45]. We confirmed the importance of this modification using a phosphomimetic NUDT5 mutant (T45D) that is unable to generate ATP in vitro and in cells, and behaves as a dominant negative in hormone regulation of gene expression [24]. We also found that the dephosphorylated NUDT5 forms a hexameric structure similar in size to that of the NMNAT1 enzyme that generated NAD+ for PAR synthesis and pyrophosphate, possibly for ATP synthesis [45]. 

### 2.2. Role of NUDT5 in Breast Cancer Stem Cells (BCSC)

As discussed earlier NUDT5 overexpression is associated with a more aggressive breast cancers, and we have provided further insight into the possible explanation for this correlation using 3D cell culture of breast cancer cells [25]. NUDT5 is essential not only the response to progesterone but of breast cancer cells to hormones, but also generation and maintenance in BCSC from multiple breast cancer cell lines grown in 3D culture [25]. In particular, and using NUDT5 mutants unable to synthesize ATP but capable of hydrolyzing ADRP, we showed that it the ATP generating activity of NUDT5 that is required for BCSC generation [16]. Analysis of the global gene expression changes occurring in 2D versus 3D cell culture reveals the use of 3D culture as a more realistic in vivo model and identifies gene signatures associated with EMT and stemness that depends on nuclear ATP synthesis by NUDT5. In addition to a gene signature associated with stemness and EMT we were able to identify tumor markers that are used in clinical trials, such as mucin 1 (MUC1; used to monitor metastasis in patients) and members of the carcinoembryonic antigen (CEA) related cell adhesion molecules (CEACAM) family. CEACAM have been used for the diagnosis and monitoring of cancer recurrence following surgery, and elevated levels of CEACAM were identified as metastatic drivers in breast cancer [49,50,51], these changes in the gene expression signature initiated by the culture of cells in 3D, demonstrate that 3D culture is a more comparable model to that of the in vivo signature. These signatures, along with angiogenesis, is characteristic of aggressive breast cancer types and depends on nuclear ATP synthesis by NUDT5. Taken together, this data shows a multifaceted role of NUDT5 in aggressive breast cancer progression (Figure 3). 

### 2.3. A Specific Antibody to the Hexameric form of NUDT5 as a Tool for Breast Cancer Stratification

The above-mentioned observations point to the importance of stratifying breast cancer in terms of their capacity to synthesize ATP from ADPR. To this end, we developed a polyclonal rabbit antibody against a synthetic peptide containing the amino acids exposed in the surface of the active dephosphorylated a hexameric form of NUDT5 [45]. We have previously shown that the shift in catalytic activity from AMP to ATP generation is mediated via a structure change in NUDT5 from a dimeric to a hexameric form. This antibody does not react with the dimeric form of phosphorylated NUDT5 in histochemistry or western blots. Using tissue microarrays from breast cancer samples, staining with this antibody correlates with clinical bad prognosis and metastasis, indicating its possible value as a selection assay for treatment of these patients with inhibitors of PARP or NUDT5. 

### 2.4. Role of NUDT Enzymes in Other Cancer Types

NUDT family members also show a significant enrichment in datasets from several other cancer types (Appendix A). Focusing on NUDT1, 2, 5, and 16 we observed that although breast cancer is the patient dataset where the majority of datasets and the most statistically significant effect on outcome is observed, the expression of these NUDT enzymes is also predictive in other cancer types (Figure 2b, Appendix A, full details are given in Appendix A). 

Indeed, it has been previously shown that the expression levels of NUDT1 (also known as MTH1) are associated with a more aggressive glioblastoma and the expression of MTH1 is essential for glioblastoma cancer stem cells [52]. MTH1 is also up-regulated in melanoma and gastric cancer [53,54]. NUDT16 is an RNA de-capping enzyme and its gene promoter is methylated in over 75% of T-cell-derived leukemia, driving downregulation compared to normal tissues [55]. In addition, knockdown of NUDT16 in Hela cells reduces cell proliferation [56]. NUDT5 was identified as an independent prognostic factor in colorectal cancer, associated with poor overall survival in clear cell renal carcinoma [57] and, in conjunction with high expression of MTH1, predicts poor survival in esophageal carcinoma [58].

As summarized above, NUDT enzymes are key players in cancer progression and may provide promising therapeutic targets for the future management of cancer patients. However, it is also clear that NUDT enzymes and their effect on the pathways in cancer are dependent on the upstream/downstream pathways, cofactors, interactors, and regulators. For example, NUDT2 is a positive regulator of mTORC [59], NUDT5 is directly dependent on the upstream activation of PARP1 and the subsequent increase in PARylation [24], and NUDT16 is required for p53 stabilization and cell survival in BRCA1 mutant cancers [60]. In addition, the positive correlation of these NUDT enzymes and their respective cofactors in cancer patient gene expression datasets (Figure 2c–e) further supports the greater understanding of the underlying pathways that may have a significant impact on the development of breast cancer. The analysis of omics data may provide further insight into these processes as discussed below.

## 3. Exploiting Global Signaling and Omic Data to Discover Novel Therapeutic Strategies 

Analysis of global gene expression and phosphoproteomic data has already revealed some interesting findings and highlighted the role of NUDT family members in cancer progression. For example, RNA-seq gene expression and subsequent network analysis in chronic myelogenous leukemia cells in culture highlighted the potential of NUDT2 as a therapeutic target, as inhibition of NUDT2 significantly affected gene pathways involved in metastasis, invasion, and apoptosis [31]. In 2015 the gene expression analysis of 30 cancer cell lines identified NUDT4 as a potential target [61]. The combination and analysis of muti-omic data revealed NUDT7 as having a role in colorectal and breast cancer development [62,63].

### Progesterone Signaling in Breast Cancer Cells

We have recently contributed to the understanding of breast cancer progression via the analysis of phosphoproteomic datasets [64]. As discussed earlier, NUDT5 generates ATP in the nucleus of breast cancer cells in response to progesterone, dependent on degradation of PARP1 synthesize PAR to ADPR by PARG. The activation of both PARP1 and NUDT5 enzymes is regulated by phosphorylation. PARP1 is rapidly phosphorylated within the NAD+ binding site following hormone, which results in a more open conformation and an increased enzymatic activity [34,35]. NUDT5 is dephosphorylated in response to progesterone which results in a change in the oligomeric structure of NUDT5, favoring a hexameric structure, which facilitates the entry of PPi and the generation of ATP [24,45]. In order to gain more insight into the mechanism of PARP1 and NUDT5 activation in breast cancer cells, a more comprehensive analysis of the phosphorylation events and signaling cascades is required. Several single events within the signaling network induced by progesterone in breast cancer cells have been studied previously, for example, the rapid activation of the MAPK cascade [65,66], but comprehensive global analysis was missing. 

To address this, we carried out a global phosphoproteomic analysis over time from antibody arrays and mass spec datasets in order to determine the signaling pathways induced by progesterone in breast cancer cells. As expected, we identified the rapid activation of the MAPK cascade, but we were also able to reconstruct activation of new kinase signaling networks previously not associated with breast cancer cell response to progesterone, including ERBB-EGF, Fc receptor, insulin, and TRK (Tropomyosin receptor kinase) signaling cascades [64]. In addition, we identified signaling networks involved in the processes of EMT, cell adhesion, and angiogenesis, which is consistent with our hypothesis and previous work and with the role of NUDT5 in aggressive cancer progression (Figure 2 [25]). Future work will focus on the interrogation of these signaling and protein-protein interaction networks to try and decipher the upstream regulators of PARP1 and NUDT5, which may provide insight into future drug combination strategies for the management of breast cancer patients, with a focus on metastatic disease for which very few therapeutic options exist. 

## 4. Conclusions

The NUDT family of enzymes is an intriguing family of enzymes that play a key role in the progression of breast and other cancers. The therapeutic potential of NUDT-selective inhibitors has already been shown—NUDT1-selective inhibitors have shown efficacy in multiple studies in vitro and in vivo [67,68,69,70], and the NUDT5-selective inhibitor TH1457 blocks cell proliferation in breast cancer cells [71]. Given the interest in NUDT inhibition for the management of cancer patients, a greater understanding of the signaling pathways impinging on the activity of NUDT enzymes is required and may be facilitated by the interrogation of global phosphoproteomic and gene expression datasets (Figure 4). 

## Figures and Tables

**Figure 1 ijms-22-02267-f001:**
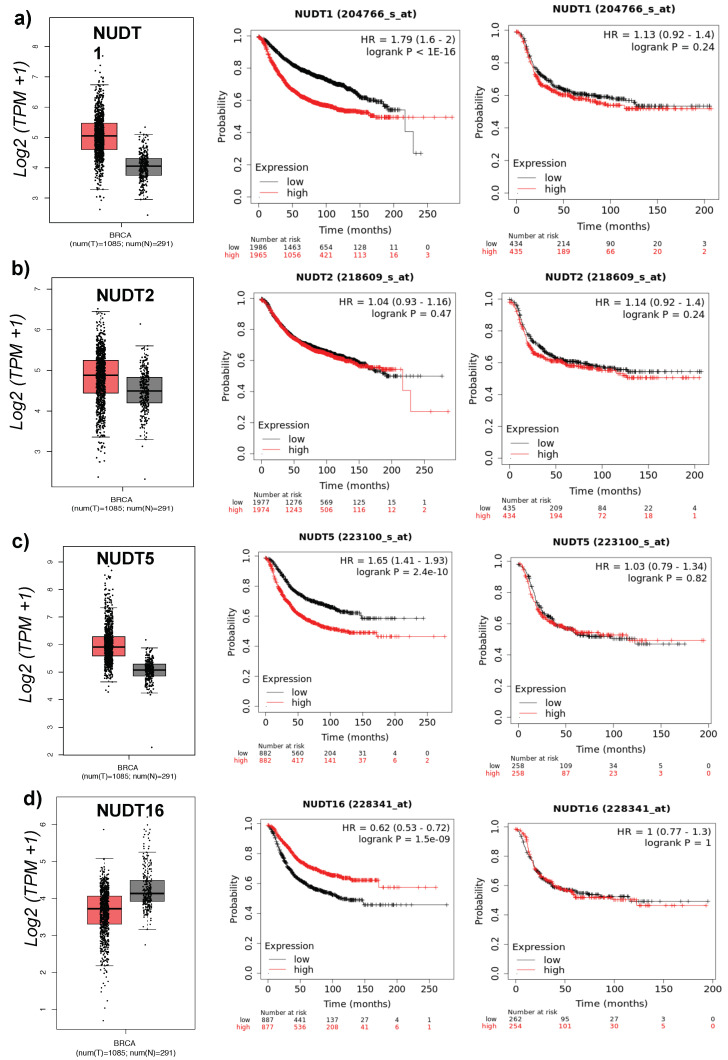
NUDT1, 2, 5, and 16 in Breast Cancer: (Left) mRNA expression levels of NUDT family member in normal (black) versus tumor (red) patient breast cancer TCGA samples. (Center) Overall survival based on patient stratification using NUDT enzyme in hormone receptor-positive breast cancer. (Right) Overall survival based on patient stratification using NUDT enzyme in hormone receptor-negative breast cancer patient datasets, (**a**) NUDT1 (**b**) NUDT2 (**c**) NUDT5, and (**d**) NUDT16.

**Figure 2 ijms-22-02267-f002:**
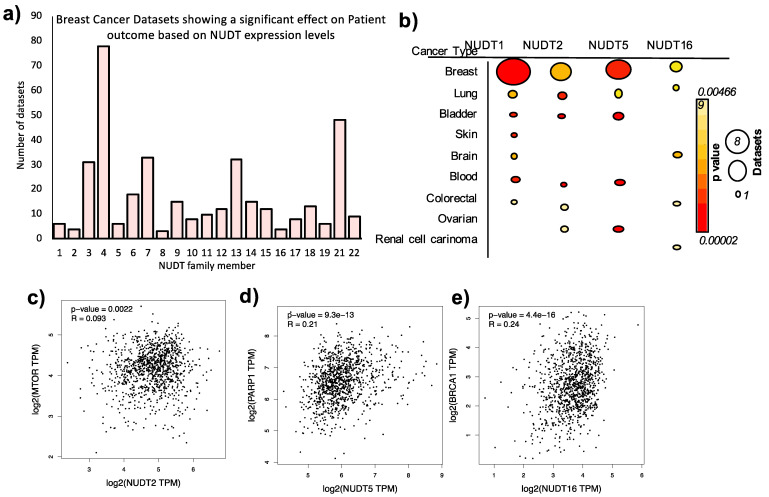
Expression Levels of NUDT enzymes in Breast and Other Cancer Types. (**a**) Number of datasets where there is a significant (*p* < 0.05) difference between overall or recurrence-free survival of patients based on the expression level of the NUDT family member indicated, full information regarding dataset ID, cohort number are given in Appendix A. (**b**) Number of datasets (indicated by size of circle) and mean *p* values (color circle) where a significant difference in cancer patient outcome is observed based on the expression level of the NUDT enzyme indicated in different types of cancer. Full details of the information are given in Appendix A. Correlation in expression of NUDT2 and MTOR (**c**), NUDT5 and PARP1 (**d**) and NUDT16 and BRCA1 (**e**).

**Figure 3 ijms-22-02267-f003:**
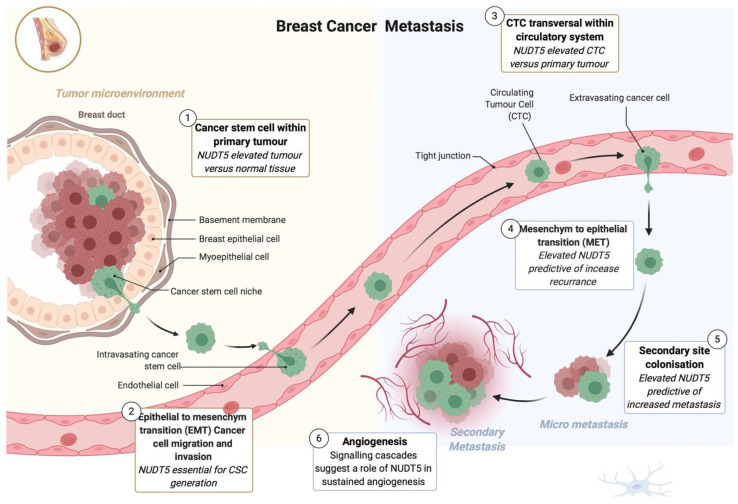
Multifaceted role of NUDT5 in breast cancer metastasis. Model showing the multiple roles and indications for a key role of NUDT5 in aggressive breast cancer. (1) NUDT5 is elevated in tumor versus normal breast cancer tissue. (2) NUDT5 is essential for breast cancer stem cell (BCSC) generation and maintenance. (3) NUDT5 is highly expressed in circulating tumor cells (CTCs). (4 and 5) Elevated levels of NUDT5 are associated with increased levels of recurrence and metastasis in patients suggesting a role in mesenchyme to epithelial transition and secondary site colonization. And finally (6) analysis of the gene expression changes occurring in BCSC in 3D cell culture suggests a role of NUDT5 in angiogenesis.

**Figure 4 ijms-22-02267-f004:**
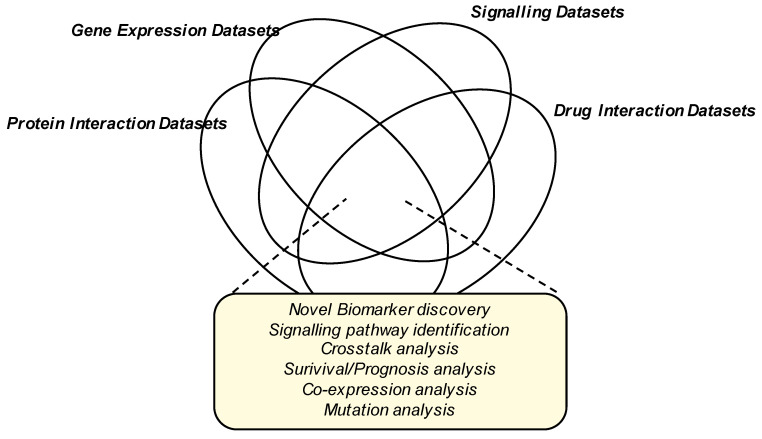
Data Integration for the identification of novel biomarkers, drug discovery targets and pathways in breast cancer progression. Venn diagram showing the possible different dataset integration strategies which may provide further insight for future cancer drug discovery lines of research.

**Table 1 ijms-22-02267-t001:** NUDT family members substrates and reactions.

NUDT(Alternative Name)	Substrate Hydrolase Activity	Product Hydrolase Activity	Role	Enzyme Entry(Expasy)
**NUDT1** (**MTH1**)	8-oxo-dGTP+ H_2_O8-oxo-dATP8-OH-dATP	8-oxo-dGMP +diphosphate+H+	Sanitizing oxidized nucleotides	EC 3.6.1.55EC 3.6.1.56
**NUDT2**(**APAH1**)	Ap4A	AMP + ATP	Homeostasis	EC 3.6.17
**NUDT3**(**DIPP-1**)	Diphospho-myo-inositol polyphosphate + H_2_O	myo-inositol polyphosphate + phosphate.		EC 3.6.1.52
**NUDT4**(**DIPP-2**)	Diphospho-myo-inositol polyphosphate + H_2_O	myo-inositol polyphosphate + phosphate.	Signal transduction	EC:3.6.1.52
**NUDT5**(**NUDIX5**)	ATP + D-ribose 5-phosphate + H+ADP-D-ribose + H_2_O8-oxo-dGDP + H_2_OADP-D-ribose + PPi + H_2_O	ADP-D-ribose + diphosphateAMP + D-ribose 5-phosphate + 2 H+8-oxo-dGMP + H+ + phosphateATP+ D-ribose-5-phosphate	Sanitizing oxidized nucleotides	EC 2.7.7.96EC 3.6.1.13EC 3.6.1.58
**NUDT6**(**FGF2AS**)	ADP-ribose + H_2_ONADH + H_2_O	AMP + D-ribose 5-phosphate.AMP + NMNH + 2 H+	Cell proliferation	EC 2.7.7.96
**NUDT7**	acetyl-CoA + H_2_O	acetate + CoA + H(+).	Eliminate oxidized coenzyme A (CoA)	EC 3.1.2.20
**NUDT8**	*unknown*	*unknown*	*unknown*	
**NUDT9**	ADP-D-ribose + H_2_O	AMP + D-ribose 5-phosphate + 2 H+		EC 3.6.1.13
**NUDT10**(**DIPP3A, APS2**)	H_2_O + P1,P6-bis(5′-adenosyl) hexaphosphateH_2_O + P1,P5-bis(5′-adenosyl) pentaphosphateDiphospho-myo-inositol polyphosphate + H_2_O	adenosine 5′-pentaphosphate + AMP + 2 H+adenosine 5′-tetraphosphate + AMP + 2 H+myo-inositol polyphosphate + phosphate	Signal transduction	EC 3.6.1.60EC 3.6.1.60EC 3.6.1.52
**NUDT**(**Alternative Name**)	Substrate Hydrolase Activity	Product Hydrolase Activity	Role	Enzyme Entry(Expasy)
**NUDT11**(**DIPP3B, APS1**)	H_2_O + P1,P6-bis(5′-adenosyl) hexaphosphateH_2_O + P1,P5-bis(5′-adenosyl) pentaphosphateDiphospho-myo-inositol polyphosphate + H_2_O	adenosine 5′-pentaphosphate + AMP + 2 H+adenosine 5′-tetraphosphate + AMP + 2 H+myo-inositol polyphosphate + phosphate	Signal transduction	EC 3.6.1.60EC 3.6.1.60EC 3.6.1.52
**NUDT12**	H_2_O + NAD+H_2_O + NADH	AMP + β-nicotinamide D-ribonucleotide + 2 H+AMP + 2 H+ + reduced β-nicotinamide D-ribonucleotide	Regulate nicotinamide	EC 3.6.1.22EC 3.6.1.22
**NUDT13**	*unknown*	*unknown*	*unknown*	
**NUDT14**(**UGPP**)	UDP-sugar + H_2_O	UMP + alpha-D-aldose 1-phosphate	*unknown*	EC:3.6.1.45
**NUDT15**(**MTH2**)	a ribonucleoside 5′-triphosphate + H_2_Oa 2′-deoxyribonucleoside 5′-triphosphate + H_2_O	a ribonucleoside 5′-phosphate + diphosphate + H+a 2′-deoxyribonucleoside 5′-phosphate + diphosphate + H+	Sanitizing oxidized nucleotides	EC:3.6.1.9
**NUDT16**	a 5′-end (N7-methyl 5′-triphosphoguanosine)-adenosine in mRNA + H_2_O	a 5′-end phospho-adenosine in mRNA + 2 H+ + N7-methylguanosine 5′-diphosphate	RNA decapping enzyme	EC 3.6.1.62
**NUDT17**	*unknown*	*unknown*	*unknown*	*unknown*
**NUDT18**(**MTH3**)	8-oxo-dGDP + H_2_O	8-oxo-dGMP + H+ + phosphate	Removes oxidized guanine from DNA and RNA	EC 3.6.1.58
**NUDT19**	acyl-CoA + H_2_O	CoA + a carboxylate.	Hydrolysis CoA esters	EC 3.1.2.20
**NUDT20**(**DCP2**)	a 5′-end (N7-methyl 5′-triphosphoguanosine)-adenosine in mRNA+ H_2_O	a 5′-end phospho-adenosine in mRNA + 2 H+ + N7-methylguanosine 5′-diphosphate	RNA decapping enzyme	EC 3.6.1.62
**NUDT21**(**CFIm25**)	*No hydrolase activity*	*No hydrolase activity*	Pre-mRNA processing

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
