# Peer review of "Role of the NUDT Enzymes in Breast Cancer"

_ijms, 2021, doi:10.3390/ijms22052267_

Round 1

Reviewer 1 Report

The review article "Role of the NUDT5 and other NUDT family members in aggressive breast cancer and possible future experimental approaches" by Wright and Eaton, provides a summary on the roles and clinical significance of NUDT protein family in breast cancer development.

I would like to congratulate the authors on their well-organized and well written  text.  

Based on their experience, they provided a comprehensive review on the field. I have learned interesting details on the roles of NUDT proteins in breast cancer. I have no additional suggestions for the authors.

Author Response

Reviewer 1: The review article "Role of the NUDT5 and other NUDT family members in aggressive breast cancer and possible future experimental approaches" by Wright and Eaton, provides a summary on the roles and clinical significance of NUDT protein family in breast cancer development.

I would like to congratulate the authors on their well-organized and well written  text.  

Based on their experience, they provided a comprehensive review on the field. I have learned interesting details on the roles of NUDT proteins in breast cancer. I have no additional suggestions for the authors.

Response: We sincerely thank the reviewer for their review and comments. We are pleased that they find the review comprehensive and interesting

Reviewer 2 Report

Authors present a review article about the role of NUDT enzyme family in breast cancer and partially in other cancers. Their record of original works shows that they are well familiar with the subject. However, I recommend some points to be clarified, and particularly to specifiy some global statements about 3D culture models, that cannot be kept up in this manner. Furthermore, the article can be more streamlined with regard to the title: Is the topic NUDT5, or all NUDT enzymes? And is it about breast cancer, or all tumors?

Figure 1, left image respectively: is this gene or protein expression?

Section 1.2: please give some more evidence that 3D culture really are a “more suitable model” for studying cancer cell growth. There are lots of 3D cell culture models, particularly scaffold-containing and scaffold-free models, all with very different characteristics, advantages and drawbacks.

Section 2.2, line 201-202: the statement 3D culture models are “more realistic” or even an “in vivo model” is too global and too superficial. Please differentiate and discuss this point in more detail.

Section 2.3: this is pretty brief- can some more background information be added for the non-expert reader?

Section 2.4, line 257: is this reference to figure 2 instead of figure 3?

English grammar and style needs revision, and spelling mistakes must be corrected.

Author Response

Reviewer 2: Authors present a review article about the role of NUDT enzyme family in breast cancer and partially in other cancers. Their record of original works shows that they are well familiar with the subject. However, I recommend some points to be clarified, and particularly to specifiy some global statements about 3D culture models, that cannot be kept up in this manner.

Response: We thank the reviewer for their comment we have included more details with regards to the use of 3D cell culture models, we have included more detail in comment below which addresses the same concern.

Reviewer 2: Furthermore, the article can be more streamlined with regard to the title: Is the topic NUDT5, or all NUDT enzymes? And is it about breast cancer, or all tumors?

 Response: We thank the reviewer for this comment regarding the title. We believe that NUDT is more appropriate than NUDT5, as detailed Table 1, and section 2.1 and 2.2 we do provide a review of the literature with regards to other enzymes not only NUDT5. In addition, we would like to, with this review draw people´s attention to the possible role of NUDT1, 2, 5 and 16 in other cancers not only breast (Figure 2b). Also, we believe that the number of patient datasets where other members of the NUDT family show a significant association with patient outcome (Fig 2a) would be of interest to those in the field. For example (and detailed in the text), NUDT4 and 21 expression is associated with patient outcome in a large number of patient datasets and there has been no investigation to date, with regards to the possible role of these enzymes in cancer progression. Given this we believe that the title should remain as NUDT and cancer in order to reach a wider audience with the data we present for other NUDT members and other cancers.

Reviewer 2: Figure 1, left image respectively: is this gene or protein expression?

Response: We apologise for any confusion. This is indeed mRNA expression (Transcripts per million TPM), we have clarified this in the figure legend.

Reviewer 2: Section 1.2: please give some more evidence that 3D culture really are a “more suitable model” for studying cancer cell growth. There are lots of 3D cell culture models, particularly scaffold-containing and scaffold-free models, all with very different characteristics, advantages and drawbacks.

Response: We thank the reviewer for this comment and they are correct there are many different ways in which 3D cell culture models can be carried out. However, a detailed discussion of the advantages and disadvantages of the use of 3D culture is not the aim of this review. This has been covered extensively elsewhere, therefore we have included and addition description of the different methods used (as suggested) and included additional references regarding 3D cell culture.

Reviewer 2: Section 2.2, line 201-202: the statement 3D culture models are “more realistic” or even an “in vivo model” is too global and too superficial. Please differentiate and discuss this point in more detail.

Response: We thank the reviewer for this comment, we agree that perhaps the comment is too global. We have now changed this to reflect, our specific 3D cell culture model the details of which we published last year. In that manuscript we were able to show using global RNA-sequencing that genes associated with EMT and stemness were increased in 3D versus 2D culture. In addition, we identified markers of breast cancer progression (for example MUCIN) which are not expressed in 2D culture but were identified in 3D. We believe that this does not provide a realistic model but a “more realistic” model than 2D cell culture based on the fact that the gene expression signature is correlated with in vivo expression signatures.

Reviewer 2: Section 2.3: this is pretty brief- can some more background information be added for the non-expert reader?

Response: We have now included some additional information to explain the racionale behind the generation of an antibody which has the possibility to be used for patient stratification. We hope that now this is acceptable.

Reviewer 2: Section 2.4, line 257: is this reference to figure 2 instead of figure 3?

Response: Indeed, the reviewer is correct we apologise for any confusion and this has now been corrected in the text.

Reviewer 2: English grammar and style needs revision, and spelling mistakes must be corrected.

Response: We thank the reviewer for their attention to detail, and we have corrected the grammar and spelling mistakes.

Round 2

Reviewer 2 Report

All my concerns were properly addressed and respective revisions were made. I recommend acceptance in the present form